# Ribosomal Hibernation-Associated Factors in *Escherichia coli*

**DOI:** 10.3390/microorganisms10010033

**Published:** 2021-12-24

**Authors:** Yasushi Maki, Hideji Yoshida

**Affiliations:** Department of Physics, Osaka Medical and Pharmaceutical University, Takatsuki 569-8686, Japan; yasushi.maki@ompu.ac.jp

**Keywords:** 100S ribosome, ribosome modulation factor, hibernation-promoting factor, ribosomal hibernation, stress response

## Abstract

Bacteria convert active 70S ribosomes to inactive 100S ribosomes to survive under various stress conditions. This state, in which the ribosome loses its translational activity, is known as ribosomal hibernation. In gammaproteobacteria such as *Escherichia coli*, ribosome modulation factor and hibernation-promoting factor are involved in forming 100S ribosomes. The expression of ribosome modulation factor is regulated by (p)ppGpp (which is induced by amino acid starvation), cAMP-CRP (which is stimulated by reduced metabolic energy), and transcription factors involved in biofilm formation. This indicates that the formation of 100S ribosomes is an important strategy for bacterial survival under various stress conditions. In recent years, the structures of 100S ribosomes from various bacteria have been reported, enhancing our understanding of the 100S ribosome. Here, we present previous findings on the 100S ribosome and related proteins and describe the stress-response pathways involved in ribosomal hibernation.

## 1. Introduction

Bacteria have developed sophisticated adaptive systems to survive during various environmental changes, including nutrient starvation, temperature shock, osmolarity changes, and rapid pH changes. These systems include inhibition of cell growth, reduction of cell volume, changes in cell shape, compression of nucleoids, changes in the cell wall composition, and changes in cytoplasmic components [1]. These environmental adaptations may result in the formation of biofilms encased in extracellular polymers [2] or the conversion of bacteria into persister cells with reduced vitality [3], making them resistant to antibiotics and responsible for chronic infections [4,5,6,7]. To further combat these stress conditions, bacteria utilize a system to dimerize and inactivate ribosomes [8]. In some gammaproteobacteria, such as *Escherichia coli* and *Vibrio cholerae*, two ribosomal protein factors, ribosome modulation factor (RMF) and hibernation-promoting factor (HPF), bind to the 70S ribosome to produce its dimeric form, the 100S ribosome. In most other bacteria, including *Staphylococcus aureus*, *Lactobacillus paracasei*, and *Thermus thermophilus*, the 100S ribosome is formed through binding of a long-form of HPF to the 70S ribosome [9,10,11]. A ribosomal hibernation model has been proposed in which the 100S ribosome loses its translational activity to reduce cellular energy consumption and functions as a reservoir for protecting ribosomes from degradation by RNases [12,13,14,15]. Recently, high-resolution structures of 100S ribosomes from various bacteria have been reported, improving the understanding of ribosome hibernation [16,17,18,19,20].

In this review, we describe previous findings on the 100S ribosome and related protein factors in *E. coli*, as shown in Table 1. We attempted to integrate the hibernation stage into the ribosome cycle and discuss the elaborate stress response pathway involved in bacterial translational control.

## 2. 100S Ribosome

Various types of ribosomal dimers have been observed using negative-staining electron microscopy since the 1970s [21,22]. However, ribosomal dimers are not always observable by electron microscopy. Some of these complexes were thought to be linked by ribosome-bound mRNAs, whereas others were considered artificial because studies using sucrose density gradient centrifugation showed that ribosomes dimerized in the presence of excess divalent cations in solution. In the 1980s, Wada et al. established an experimental method for stably obtaining ribosomal dimers [23] by avoiding the washing of *E. coli* cells harvested during the stationary growth phase with fresh buffer. The stable ribosomal dimer, whose sedimentation coefficient was 100S, was analyzed using sucrose density gradient centrifugation and named the 100S ribosome. Subsequently, the protein RMF was found to bind to this ribosome, and it was confirmed that adding RMF to the ribosome in vitro caused dimerization (Figure 1) [12,23]. The addition of RMF to ribosomes in vitro resulted in a marked decrease in translational activity, indicating that the 100S ribosome is inactive [12].

100S ribosome formation is thought to inhibit the synthesis of proteins that consume large amounts of energy under conditions unfavorable for bacterial growth [8,24]. According to the results of sucrose density gradient centrifugation, around 40% of all ribosomes are converted to 100S ribosomes during the stationary growth phase [23]. However, this measured value is uncertain because the 100S ribosome of *E. coli* easily dissociates into 70S ribosomes at low ribosomal concentrations [25]. Cryo-electron microscopy tomography in situ revealed that 20–25% of ribosomes were in the 100S form; however, the culture conditions were not optimized to observe large numbers of these ribosomes [26]. In contrast, stoichiometric analysis of ribosomal binding proteins revealed that RMF binds to up to 80% or more of ribosomes in *E. coli* cells [27]. Therefore, most ribosomes are thought to be converted to 100S ribosomes in the cell depending on the environment.

In *E. coli*, ribosomes are degraded by exoribonucleases such as RNase R and RNase II in response to environmental stress [14,15]. However, it has been speculated that the 100S ribosome complex protects ribosomes from degradation during osmotic or heat stress [28,29]. Dormant cells known as persisters, which are resistant to antimicrobial drugs, contain high levels of 100S ribosomes [30]. When cells are removed from stress conditions, inactive 100S ribosomes quickly dissociate into active 70S ribosomes and protein synthesis resumes [31]. It has been reported that EF-G/RRF or Hflx is involved in the dissociation of 100S ribosome in *S. aureus* [32]. However, the 100S ribosome of *S. aureus* is formed by the long form of HPF and not by RMF [10]. Although EF-G/RRF and Hflx are also present in *E. coli*, it has not been confirmed whether these protein factors can dissociate the 100S ribosome formed by RMF in *E. coli*. In *E. coli*, it has been reported that IF3 releases HPF from ribosomes but does not release RMF [33].

These studies suggest that the interconversion system between active 70S ribosomes and inactive 100S ribosomes is an important strategy for cells to survive in harsh environments. This stage in the translational cycle, in which the ribosome has lost its activity, is named the hibernation stage, and ribosomes in this stage are known as hibernation ribosomes [12].

Information on the detailed structure of the 100S ribosome in Gram-negative organisms is limited. The structure of the *E. coli* 100S ribosome was revealed using cryo-electron microscopy in 2010 [25]. The 100S ribosome is formed by two 70S ribosomes that contact each other at the 30S subunit (see Figure 1). At the junction of the 30S subunits, the ribosomal protein S2 of one subunit fits into the pocket formed by the ribosomal proteins S3, S4, and S5 of the other subunit. This pocket is a pathway for mRNA; therefore, mRNA cannot bind to the 100S ribosome if this pocket is blocked by ribosome dimerization. In addition, an extra dense mass has been observed in the 30S subunit of the 100S ribosome, which was not present in the structure of the previously observed 70S ribosome. These structural features of the 100S ribosome have also been observed by cryo-electron tomography in situ [26]. Recent structures determined that using cryo-electron microscopy with an improved resolution visualized RMF and HPF on the *E. coli* 100S ribosome, as shown in Figure 1 [18]. This structure revealed that RMF and HPF mediate 70S dimerization indirectly by stabilizing the ribosomal proteins S1 and S2, in contrast to those observed in Gram-positive bacteria in which the long-form of HPF directly participates in the dimerization interface [16,17,19,20,34]. The structure also indicated the presence of deacylated E-site tRNA and ribosomal protein S1 on the 100S ribosome. This ribosomal protein S1 is thought to be the extra-dense mass mentioned above. Interestingly, S1 has an inactive structure for translation initiation and contacts the RMF [18].

## 3. Preparation of Ribosomal Hibernation

### 3.1. Synthesis of (p)ppGpp on the Ribosome by RelA

Expression of the *rmf* gene is known to be controlled by guanosine tetraphosphate and pentaphosphate, collectively known as (p)ppGpp [35]. Depletion of essential nutrients in the environment is among the most serious threats to bacteria. A central component of adaptation to this stress is the stringent response [36] driven by (p)ppGpp, which plays numerous roles in regulating cell growth rates and adapting to the environment [37]. (p)ppGpp regulates transcriptional activity and decreases ribosome biosynthesis for environmental adaptation by interacting with RNA polymerase [38,39]. Intracellular (p)ppGpp levels are regulated by enzymes belonging to the RelA/SpoT homolog family [40]. SpoT is a bifunctional enzyme with strong hydrolytic activity and weak (p)ppGpp synthesis activity (<8> in Figure 2) [41], whereas RelA is a ribosomal factor with (p)ppGpp synthesis activity (see Table 1) [42]. When the supply of amino acids is limited by nutrient starvation, deacylated tRNA binds to the A site of the 70S ribosome (<6> in Figure 2). RelA is activated by binding to deacetylated tRNA-bound 70S ribosomes [43]. Activated RelA transfers the pyrophosphoryl group from ATP to GTP or GDP on the ribosome, synthesizing pppGpp or ppGpp, respectively (<7> in Figure 2). To understand how RelA synthesizes (p)ppGpp on the deacylated tRNA-bound 70S ribosome, the high-resolution structure of the entire 70S RelA deacyl-tRNA complex was analyzed [44,45,46]. The structural analysis results showed that the interaction between RelA and the ribosome caused conformational changes in both the 30S and 50S subunits during (p)ppGpp synthesis. (p)ppGpp is involved in promoting transcription of the *rmf* gene and strongly affects 100S ribosome formation (<9> in Figure 2) [35]. This sequence of events (amino acid starvation → synthesis of (p)ppGpp by RelA → expression of RMF by (p)ppGpp → formation of 100S ribosome by RMF → reduction of translational activity by 100S ribosome) is congruent with the response to starvation stress.

### 3.2. Changes in Ribosomes Because of Growth Phase Transition

The established ribosome cycle in bacteria consists of initiation, elongation, termination, and recycling stages (<1>–<4> in Figure 2). In the exponential growth phase, when nutrients are sufficient, ribosomes dissociate into subunits in a GTP-dependent reaction involving ribosome recycling factor and elongation factor G after the recycling phase. Thereafter, initiation factor 3 (IF3) binds to the 30S subunit to stabilize the dissociation for the next round of translation (<5> in Figure 2) [47]. However, ribosomes are less likely to be bound by IF3 as the growth phase transitions from the exponential phase to the stationary phase [48]. In contrast, when RMF is added to ribosomes prepared from cells during the exponential phase in vitro, 100S ribosomes are less likely to form than when they are prepared from cells during the stationary phase [48]. These results indicate that ribosomes in the exponential and stationary phases are different. Structural analysis showed that the ribosomal protein S1 binds to 100S ribosomes, which contacts the RMF [18]. It is known that this ribosomal protein facilitates the unwinding and placement of the start codon of mRNA [49]. Interestingly, the S1 protein on the 100S ribosome has an inactive and compact conformation that differs from previously resolved structures. Moreover, the S1 protein has been reported to interact with IF3 [33,50]. The ribosomal protein S1 may play a role in regulating the binding of IF3 to ribosomes and initiating translation or RMF binding to ribosomes and dimerizing them during the transition from the exponential phase to the stationary phase.

## 4. Factors Affecting 100S Ribosome Formation

### 4.1. Ribosome Modulation Factor

RMF in *E. coli* is a small, basic protein consisting of 55 amino acids whose main function is forming the 100S ribosome by binding to the 70S ribosome (see Table 1) (<12> in Figure 2). The *rmf* gene is mainly carried by enteropathogenic bacteria belonging to the gammaproteobacteria family, such as *E. coli*, *Vibrio cholerae*, and *Yersinia pestis* [8]. Under culture conditions in a rich medium, RMF was not expressed during the exponential growth phase, but instead, its expression was initiated when the culture entered the stationary phase. The half-life of *rmf* mRNA transcribed in the stationary phase is approximately 24 min, which is clearly longer than the typical half-life of *E. coli* mRNA (2–4 min) [31], and its expression level is mainly regulated by ppGpp (<9> in Figure 2) [35]. Surprisingly, the expression of RMF is not dependent on RpoS (sigma S), the major alternative transcriptional factor in the stress and stationary phase in *E. coli* [35]. *rmf* gene expression is reportedly regulated by numerous elements: cyclic AMP (cAMP) and cAMP receptor protein (CRP) (which are stimulated by a decrease in metabolic energy) (<10> in Figure 2); the transcription factor ArcA (which is involved in redox regulation under anoxic conditions); and the transcription factors McbR, RcdA, SdiA, and SlyA (which are involved in biofilm formation) (<11> in Figure 2) [51,52]. Cells in a biofilm may develop antibiotic resistance by forming 100S ribosomes with reduced translational activity [53]. RMF is not only involved in the formation of ribosomal dimers, but also in the resistance of cells to osmotic stress [28], heat stress [29], and acid stress [54] by limiting rRNA damage [14,15]. This indicates that RMF expression is controlled by a number of transcription factors that respond to various stresses [51].

The binding sites of RMF on the ribosome have been investigated by protein–protein crosslinking [13], chemical probing [55], and structural analyses [18,56]. The results of these structural analyses indicated that RMF binds near the anti-Shine-Dalgarno region of the 30S subunit (see Figure 1). The structure of the RMF consists of two helices connected by a linker, as determined in nuclear magnetic resonance analysis (PDB ID: 2jRM), X-ray crystallography (PDB ID: 4V8G), and cryo-electron microscopy (PDB ID: 6H4N). Thirteen amino acids of RMF are completely conserved and functionally important, as replacement of one of these residues with alanine suppressed ribosome dimerization [57]. Examination of the functional sites using mutants of RMF revealed that R3, K5, and R11 in the N-terminal domain contribute significantly to ribosome binding. Analysis of the 100S ribosome by cryo-electron microscopy showed that R3 and K5 interact with helix 28 of 16S rRNA [18]. In addition, G23 and R45 are important for ribosome dimerization because although RMFs with mutations in these amino acids bind to ribosomes, they cannot form dimers [57]. Thus, RMF contains multiple functional sites for ribosome binding and ribosome dimerization.

### 4.2. RaiA and HPF

Protein Y, the gene product of *yfiA*, was identified as a factor binding to the inter-subunit position of the 30S, which stabilizes the 70S monomer of *E. coli* ribosomes under low Mg^2+^ conditions in vitro [58]. Subsequently, this protein was found to inhibit cell-free translation of mRNA in cell extracts of *E. coli* and be induced under cold-shock stress. Thus, protein Y was renamed as ribosome-associated inhibitor A (RaiA) and its gene name (*yfiA*) as *raiA*, as shown in Table 1 [59]. It has also been reported that RaiA reduces translation errors [60].

RaiA and its paralogous protein YhbH were found to be associated with hibernating ribosomes in the stationary phase of *E. coli* cells and were released from translating ribosomes soon after starved cells were transferred into a fresh medium. This indicates that these proteins are involved in the storage form of ribosomes in the stationary phase, at least in *E. coli*, as is the case with RMF [61]. YhbH appeared to promote the formation of 100S ribosomes by converting the immature 90S dimer containing RMF; hence, the protein and its gene name were renamed as HPF and *hpf*, respectively, as shown in Table 1 (<14> in Figure 2) [62]. Although both RaiA and HPF are encoded in the *E. coli* genome and are highly homologous to each other, HPF binds to 100S ribosomes with RMF in contrast to RaiA binding to the 70S monomer in the ribosomal fraction (<15> in Figure 2) [61].

Crystallographic and chemical probing analyses revealed that RaiA blocks the peptidyl-tRNA site (P site) and aminoacyl-tRNA site (A site) of the ribosome and inhibits translation initiation [63]. It was suggested that the HPF binding site overlaps with RaiA because of the high identity of these sequences and their similar structures in solution [64]; nevertheless, HPF promotes 100S formation with RMF, in contrast to RaiA, which prevents dimer formation [62]. The 3D structure of HPF and YfiA on the ribosome showed that these proteins are bound in the channel between the head and body of the 30S subunit where tRNAs and mRNA bind during protein synthesis [56]. It was also suggested that RaiA inhibits 100S formation using its C-terminal short tail, which blocks binding of RMF and prevents RMF-induced dimer formation. Recently, a high-resolution structure of hibernating 100S ribosomes from *E. coli* was reported with well-resolved electron densities for HPF and RMF, revealing a direct interaction of HPF with E-site tRNA, as shown in Figure 1 [18].

Some information on the expression and regulatory mechanisms of the genes of RaiA and HPF in *E. coli* is known. Expression of both proteins is induced under starved conditions [61], including nitrogen starvation [65]. It has also been reported that the mRNA abundance of *hpf* is increased by the signal of autoinducer 2 (<16> in Figure 2) [66]. RaiA was induced under cold-shock treatment [59] and downregulated by FNR, a global regulator of anaerobic metabolism [67]. RaiA and RMF expression is induced by cAMP in response to glucose starvation (<17>, <10> in Figure 2) [51] and by starvation alarmone (p)ppGpp (<18>, <9> in Figure 2) [68].

Recently, some results concerning functions of these proteins and storage ribosomes have been reported in several bacterial species. The HPF- or RaiA-bound storage ribosomes of *E. coli* exhibit resistance to unfolded protein-mediated subunit dissociation and subsequent degradation by cellular ribonucleases, with the intrinsic chaperon activity retained to assist in protein folding [69]. The absence of HPF results in the loss of some proteins from the ribosome during incubation in the stationary phase [70,71]; furthermore, HPF protects ribosomes against degradation in the absence of mRNA by blocking the attack of ribonuclease [72,73,74]. Although these results include cases of specific species except for gammaproteobacteria, the inter-subunit proteins of non-active ribosomes may typically protect the machinery from degradation under stress conditions.

### 4.3. YqjD

YqjD, whose physiological function was unknown, has been reported to be involved in biofilm formation [75]. This protein is not expressed when *rpoS* is deleted, indicating that its expression is regulated by the sigma factor RpoS, which is responsible for the transcription of stationary-phase specific genes. YqjD is a membrane-binding protein with a transmembrane helix in the C-terminal region and associates with 70S and 100S ribosomes at the N-terminal region (see Table 1) [27]. Therefore, it was concluded that YqjD anchors some ribosomes to the membrane during the stationary phase (<19> in Figure 2). *yqjD* has been found only in closely related species of *E. coli*, such as *Salmonella typhimuriuma* and *Shigella flexneri*. *Escherichia coli* possesses two paralogous proteins of YqjD, ElaB, and YgaM. These paralogous proteins have transmembrane helix and ribosome binding activity, which are expressed during the stationary phase as observed for YqjD [27]. One cell has three paralogous proteins, YqjD, ElaB, and YgaM, which may be an important strategy for localizing part of the stationary-phase ribosomes to the membrane.

## 5. Conclusions

The ribosome cycle in bacteria consists of initiation, elongation, termination, and recycling stages, as shown in Figure 2. During the exponential growth phase, the ribosome progresses step-by-step in this cycle (<1> → <2> → <3> → <4> → <5> →<1> in Figure 2), during which proteins are synthesized. However, the protein synthesis activity in bacteria is inhibited under stressful conditions, such as amino acid starvation. Upon amino acid starvation, the number of deacylated tRNAs initially increases; these tRNAs bind to the A site of the ribosome (<6> in Figure 2). Once bound, RelA binds to the ribosome and initiates the synthesis of (p)ppGpp (<7> in Figure 2). Under stress conditions, such as fatty acid starvation, carbon source starvation, phosphate starvation, and hyperosmotic shock, SpoT synthesizes (p)ppGpp (<8> in Figure 2). Transcription of the *rmf* gene is mainly induced by (p)ppGpp, which is synthesized by RelA and/or SpoT (<9> in Figure 2) and regulated by the cAMP-activated global transcriptional regulator CRP (<10> in Figure 2) and the transcription factors involved in biofilm formation McbR, RcdA, SdiA, and SlyA (<11> in Figure 2).

RMF, translated as a protein factor, competes with IF3 for binding to ribosomes that have completed the recycling step (determining whether the ribosome proceeds to <5> or <12> in Figure 2). Prior to this event, as the *E. coli* culture transitions from the exponential phase to the stationary phase, changes to the ribosome make it less likely to bind to IF3 and more likely to form the 100S ribosome mediated by RMF. As ribosomal proteins, S1 and IF3 are bound to the 30S subunit in the initiation stage and inactive S1 and RMF are bound to the 100S ribosome, S1 protein may be involved in the ribosomal changes described above. The binding of RMF results in the formation of a 70S dimer, which is recognized as a particle with a sedimentation coefficient of 90S in sucrose density gradient centrifugation (<13> in Figure 2). The binding of HPF to the unstable 90S particle transforms it into a mature 100S ribosome (<14> in Figure 2). HPF expression is known to be controlled by autoinducer 2. In contrast, RaiA, a paralog of HPF, suppresses the formation of the 100S ribosome by binding to the 70S ribosome (<15> in Figure 2). Its expression is under the control of (p)ppGpp (<18> in Figure 2) and cAMP-CRP (<17> in Figure 2) as well as RMF. Some of these 70S and 100S ribosomes, which have lost their translation activity during the stationary phase, are localized near the membrane by YqjD (<19> in Figure 2). When the surrounding environment of the cell is improved, the 100S ribosome is immediately converted to an active 70S ribosome by releasing RMF and HPF, and the ribosome cycle resumes (<20> in Figure 2). This ribosomal hibernation system is thought to be suitable for the life cycle of enterobacteria, which are released from starvation and multiply at once when their host eats.

The genes for these ribosomal hibernation factors are scattered throughout the *E. coli* genome (see Table 1) and use different expression pathways (see Figure 2). The expression levels of these proteins may be regulated according to the type of stress. The formation of 100S ribosomes is a key strategy for bacterial survival under various stress conditions, which protects bacteria in the presence of antibiotics. Therefore, determining the expression mechanism of protein factors, such as RMF and HPF, which contribute to the formation of 100S ribosomes, will be useful for considering the effective use of antibiotics.

## Figures and Tables

**Figure 1 microorganisms-10-00033-f001:**
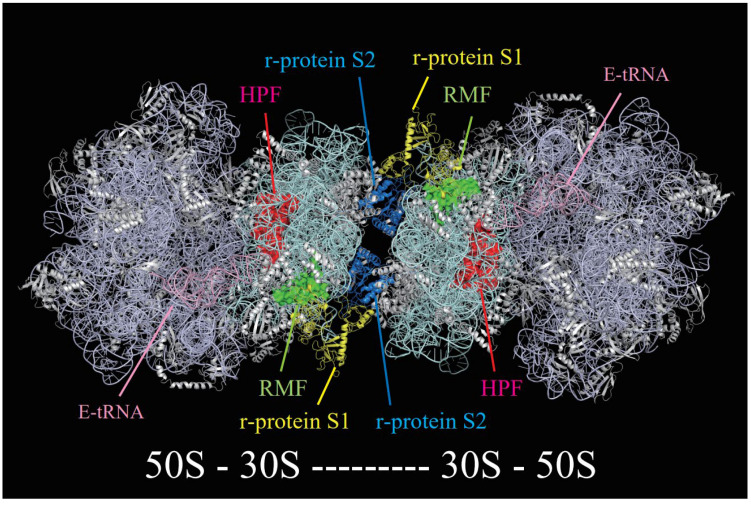
Structure of the 100S ribosome in *Escherichia coli* (PDB ID: 6H58). The 100S ribosome is formed by 30S subunits of the two 70S ribosomes (50S-30S-30S-50S). RMF, the key factor for ribosome dimerization, is shown in green. HPF, a factor that stabilizes the 100S ribosome, is shown in red. Ribosomal proteins S1 and S2, which play important roles in forming the 100S ribosome, are shown in yellow and blue, respectively. The tRNA of the E site bound to the 100S ribosome is shown in pink.

**Figure 2 microorganisms-10-00033-f002:**
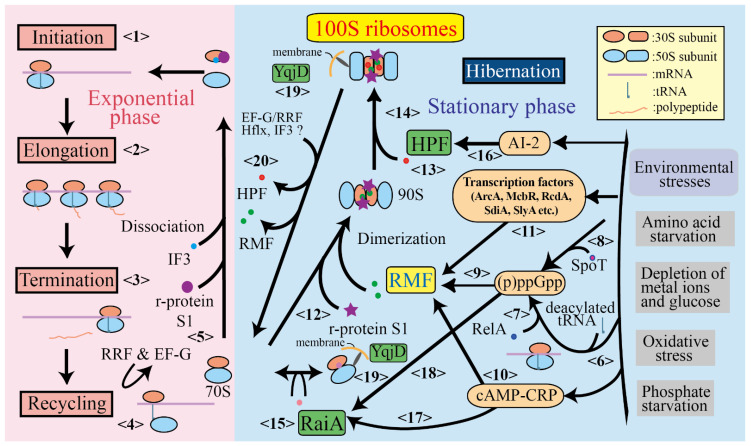
Ribosome cycle including the hibernation stage and expression pathways of factors related to formation of 100S ribosome in *Escherichia coli*. The state of ribosomes in the exponential phase, when cell proliferation is active, and in the stationary phase, when cells are under stress, are shown. The currently known expression pathways of RMF, HPF, and RaiA, factors involved in the formation of the 100S ribosome, are depicted.

**Table 1 microorganisms-10-00033-t001:** Factors related to ribosome hibernation in *Escherichia coli*.

Name	M*r*	Gene Position	Main Function Related to 100S Ribosomes
**Factors directly involved in 100S ribosome formation**			
RMF	6.5 k	1016137–1016304	Dimerizing ribosomes
HPF	10.8 k	3346028–3346315	Stabilizing 100S ribosomes
**Major factors indirectly involved in 100S ribosome formation**			
RaiA	12.8 k	2735810–2736151	Inhibiting ribosome dimerization
RelA	83.9 k	2910073–2912307	Synthesis of (p)ppGpp
YqjD	11.1 k	3248031–3248336	Localizing ribosomes to membrane

## Data Availability

Not applicable.

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
