# Peer review of "Ribosomal Hibernation-Associated Factors in Escherichia coli"

_microorganisms, 2021, doi:10.3390/microorganisms10010033_

Round 1

Reviewer 1 Report

Dear Authors
Thank you very much for your manuscript submission. The topic of your manuscript seems interesting. Your manuscript is well-designed; however:

1)  You can add some sections regarding the association of inactive 100S and and active 70S ribosomes with ESBL- and metallo-beta-lactamase production and the related mechanisms and factors in E.coli pathotypes. The association of inactive 100S and and active 70S ribosomes with multi-drug resistance characteristics and the related mechanisms and factors in E.coli pathotypes . In this regard, you can read and add the following papers to the manuscript References section.

Antimicrobial Agents and Urinary Tract Infections. Curr Pharm Des. 2019;25(12):1409-1423. doi: 10.2174/1381612825999190619130216. PMID: 31218955.

Metallo-ß-lactamases: a review. Mol Biol Rep. 2020 Aug;47(8):6281-6294. doi: 10.1007/s11033-020-05651-9. Epub 2020 Jul 11. PMID: 32654052.

2) The use of tables and figures may support your manuscript to be more comprehensible and influential.

3) The used references are too old in many cases. So, a serious refresh is needed for References section.

4)  Due to aforementioned items, a deep major revision is needed.

Author Response

We appreciate your insightful comments on our paper, which helped us to improve the manuscript. In response to your suggestion, we have added an explanation of the relationship between the figures in the text. Corrected sections in the paper are indicated in red text with comments. Recently, many papers have described 100S ribosomes mediated by long-HPF, whereas RMF, which is present in only some bacteria, has not been widely reported. Therefore, it is inevitable that there are many older reference papers. (Some recent papers on long-HPF-mediated 100S ribosomes have been added as reference papers.) Thank you for your advice.

Reviewer 2 Report

Dear Editor,

Maki et al have presented a comprehensive review on the formation of 100S ribosomes and have described the proteins that actively promote this dimer formation. The authors also elaborate on factors and environmental conditions that promote the ribosome hibernation process. The authors finally conclude by providing an overview of the ribosome cycle from protein synthesis to recycling.

The authors have performed a clear and comprehensive review of the literature and presented it with clarity.  I do have a few minor suggestions that the authors could briefly elaborate on. These are outlined below.

It would be useful to state how are 100S ribosomes are recycled and what factors are involved? There was a recent study by Basu et al JBC 2020 which would perhaps be useful to mention along with the description of factors from other bacteria.

Line 82-83: Please state briefly in the text the role of HPF in 100S ribosome structure.

Lines 128-133: It would be useful to state how S1 ribosomal protein could play a role in RMF binding.

Figure 1 legend: Please correct 100S ribosomes are formed by ‘Figure’ 30 S subunit.

Author Response

We appreciate your insightful comments on our paper, which helped us improve the manuscript. Although there are reports of long-HPF-mediated recycling pathways of 100S ribosomes, it has not been directly demonstrated that the same recycling factors are involved in the recycling pathway of E. coli 100S ribosomes because the formation type (mediated by long-HPF or RMF) is different. This information has been added to the text. Corrected sections in the paper are indicated in red text with comments. Thank you for your advice.

Round 2

Reviewer 1 Report

Dear Authors

The related revisions are done rigorously. Accept